# An Experimental Study on Non-Destructive Evaluation of the Mechanical Characteristics of a Sustainable Concrete Incorporating Industrial Waste

**DOI:** 10.3390/ma15207346

**Published:** 2022-10-20

**Authors:** Tariq Umar, Muhammad Yousaf, Muhammad Akbar, Nadeem Abbas, Zahoor Hussain, Wajahat Sammer Ansari

**Affiliations:** 1Architecture and the Built Environment, University of the West of England, Bristol BS16 1QY, UK or; 2Department of Civil Engineering, University of Engineering and Technology, Lahore 39161, Pakistan; 3Institute of Mountain Hazards and Environment, Chinese Academy of Sciences, Chengdu 610041, China; 4Department of Civil Engineering, Disaster Mitigation for Structures, Tongji University, Shanghai 200070, China; 5Department of Civil Engineering, Sir Syed University of Engineering and Technology, Karachi 75300, Pakistan; 6School of Civil Engineering, Dalian University of Technology, Dalian 116024, China

**Keywords:** concrete, industrial waste, mechanical properties, non-destructive testing, ANOVA, SEM, CBA, WGS

## Abstract

Structural materials sustainability is gaining popularity across the globe at present. Reusing natural resources, building, demolition debris, and solid waste are the most apparent tools to make construction more environmentally friendly. Traditional concrete is believed to be less durable, stronger, environmentally friendly, and socially and commercially feasible than industrial waste concrete. The evolution of non-destructive testing (NDT) across time has not been investigated in depth by researchers. An experimental study was carried out to propose the use of non-destructive mechanisms that would enable us to assess concrete’s compressive strength without causing destruction. Varying quantities of industrial waste (coal bottom ash (CBA) and waste glass sludge (WGS)) were incorporated to cast concrete prisms (150 mm × 150 mm × 150 mm). The results obtained helped us to establish relationships between the compressive strength of concrete and the Schmidt hammer rebound value, as well as the ultrasonic pulse velocities. Microstructural analysis showed that incorporating 10% of CBA and WGS improved the porosity of concrete specimens, which shows the applicability of these industrial wastes as partial cement replacements. Scanning electron microscopy (SEM) showed traces of calcium alumino-silicate hydrate (C-A-S-H), portlandite and C-S-H, which indicates the binder characteristics of CBA and WGS. The concept of the response surface approach (RSM) for optimizing cement and industrial waste substitution was validated by the polynomial work expectation. The model was statistically significant when the fluctuation of ANOVA was analyzed using a *p* value with a significance level of 0.05. The study results show that the usage of 15% CBA and 10% WGS as a cementitious additive and cement replacement has the potential to increase the strength of concrete significantly.

## 1. Introduction

Concrete is one of the most often and extensively employed building materials. Structures are built around it because of its robustness and long-term viability in retaining structures such as walls and channels. Concrete is produced from a variety of materials, including cement. In addition to these fundamentals, additional admixtures are used in the mix to give concrete its diverse qualities. The demand for cement is rising due to a rise in the demand for concrete. The following gases are released as a by-product of the manufacturing process for cement: CO_2_, nitrogen and sulfur dioxide. CO_2_ constitutes the majority of the planet-warming greenhouse gases, accounting for 76% of the total [1]. Different industrial by-products are being used as fractional substitutions for cement in concrete to reduce CO_2_ emissions. Landfilling with multiple waste products is also causing an increase in ecological contamination [2]. This could be solved by the construction industry’s potential use of a wide range of waste materials [3,4,5], resulting in more effective waste management [6,7,8].

Carbon dioxide gas (CO_2_) is released into the atmosphere due to an increase in the demand for cement. Various industrial wastes or by-products are being used in concrete as a partial substitute for cement in order to reduce the CO_2_ emissions from cement plants. Coal bottom ash and waste glass sludge are two of the world’s most common waste products. Using these waste products in landfills results in serious environmental problems, such as increased soil alkalinity, polluted ground or surface water and negative effects on plants and other living organisms. Aside from helping to reduce CO_2_ emissions, the use of these waste materials as a cement substitute will also have a positive impact on the environmental hazards posed by landfilling [9,10]. Mechanical properties, including durability properties such as water permeability and chloride migration, improve significantly with the use of industrial wastes such as marble powder and carbon black [10,11,12]. Tayeh et al. [13] studied the influence of rubberized concrete, which is an industrial waste, to improve the serviceability and durability properties [14,15,16,17].

Waste materials were successfully incorporated into concrete in this study, resulting in significant societal and economic benefits. Coal-fired thermal power stations produce coal bottom ash (CBA), sometimes known as bottom ash. These plants regularly produce a large amount of CBA (10–20 percent of the aggregate coal fiery debris). The dangerous elements in CBA can travel and contaminate ground or surface water, posing a threat to plants and other living organisms when it is disposed of as landfill. Researchers began working with CBA as a partial substitute for cement, and it was revealed that the construction sector has a great deal of potential to use CBA as a cement replacement material (CRM). In addition to silica, iron and alumina, CBA contains sulfate, magnesium, calcium and other minerals [18,19]. Researchers have used different nano-sized materials as fillers and binders to study their influence and applicability in concrete [20,21,22,23].

Due to its chemical composition, CBA is a better cement substitute. In the past, researchers have used CBA as a partial substitute for cement and obtained promising results. Waste glass sludge (WGS) is a by-product of the glass cutting and finishing process that is utilized in the industrial sector. Currently, the great majority of WGS is being disposed of in landfill, which is causing a variety of environmental problems. The particle size, surface area and the overall amount of Al2O_3_ and SiO_2_ in the admixture all affect the material’s physical qualities. WGS has a high concentration of Al2O_3_ and SiO_2_ and large-surface-area particles. As a result, it may be reasonable to use WGS for cement in concrete to improve its mechanical properties. Concrete is subjected to a variety of tests, some destructive, some semi-destructive and some non-destructive [24,25].

Non-destructive testing (NDT) is primarily concerned with determining the quality and long-term performance of a concrete construction. A number of countries have embraced non-destructive testing (NDT) [26] since it does not alter the structure’s appearance, quality or performance. For the evaluation of structural conditions, ultrasonic pulse velocity (UPV) is a non-destructive test that is commonly employed. Concrete is used to propagate pulses at a specific frequency, which is the basis for this technique [27,28,29]. The length of time required for the pulse to travel through a given material is recorded. It is possible to determine the normal pulse speed once the gap and duration are illustrated, which can depend on a variety of elements, including the material’s characteristics and the amount of water within the pores, among others. An ultrasonic wave propagation variance study is widely used to identify uniform and non-uniform zones in concrete [30].

This method can also be used to identify concrete flaws. The UPV method makes it possible to continuously analyze concrete’s conditions throughout its useful life. Non-destructive testing methods include the rebound hammer test. During the test, the tester’s plunger retracts against a spring, and this spring is mechanically released when it is fully tensioned, which causes the hammer mass to contact the concrete. Members and concrete qualities (such binder type, binder content and mixture type) all play a role [31,32].

There are numerous research studies on the relationship between the number of rebounders and the concrete compressive strength. Concrete including industrial waste, on the other hand, displays unity in that there is no clear relationship between the two. According to the findings of this investigation, there are numerous advantages to using non-destructive testing (NDT) methods for concrete structures, such as the fact that no structural component is damaged during NDT, and NDT procedures are less time-consuming than destructive testing techniques [33]. Research on the interrelationships between NDT and destructive testing for concrete containing waste elements is quite scarce. The current study focuses on NDT methodologies, as well as their relationships with concrete’s qualities when integrated with industrial waste, and how these features are linked to NDT results. Different percentages of industrial waste were used to explore the correlations between the compressive strength and rebound number and ultrasonic pulse velocity. An innovative solution to the problem of estimating concrete’s qualities using NDT is presented in this study. Moreover, the response surface approach (RSM) for optimization is used to obtain a prediction of the concrete properties. The optimization outcomes are reliable and results are similar to experimental results. In addition, to demonstrate its usefulness, the studied method is used to make predictions regarding the use of CBA and WGS.

### Research Significance

Unlike previous research, our study adds to the literature by investigating different results using the RSM technique. Different industrial wastes, such as CBA and WGS, which pose a threat to the global environment, are incorporated to minimize the usage of cement. Test results are evaluated using both destructive and non-destructive techniques. Moreover, a microstructural analysis was carried out to study the C-S-H, portlandite and other cementitious properties of CBA and WGS.

## 2. Materials and Methods

In order to carry out a series of laboratory tests on a specimen, the tests for determining the physical properties of the aggregates and binders to be employed, the workability of the fresh mixes (changing % replacement values), compressive tests and non-destructive testing on the hardened mixes (rebound hammer and UPV) are required. Specific gravity and density are also considered in early testing. The current ASTM requirements were followed in the preparation of the concrete mix, preliminary testing of the material and the casting and testing of prism samples. The chemical properties of the cement, CBA and WGS are shown in Table 1.

### 2.1. Materials and Mix Proportioning

Ordinary Portland cement was used in this experimental campaign. The chemical composition of cement is shown in Table 1. CBA and WGS were also incorporated as a binder material. The detailed composition is shown in Table 1. Fine aggregates having a fineness modulus of 2.7 were used, while a coarse aggregate with a fineness modulus of 4.3 was incorporated. Properties of the used aggregates are shown in Table 2. A polycarboxylate-based superplasticizer was used as a concrete admixture to maintain the workability.

Concrete constituents were mixed in a rotating pan-type mixer for approximately ten minutes at an average speed of 32 rpm, until uniformity was attained in the concrete matrix. A steel prism of 150 mm breadth, 150 mm length and 150 mm height, and prisms of 400 mm length and 100 mm cross-section, were filled with fresh concrete in approximately three equal strata and thoroughly compacted through a vibrating table. Mixing and casting were performed in accordance with the recommendations provided in ACI 211.1. Calculation for the concrete mix design was performed, and initially a mix proportion of 1: 1.22: 2.7 was selected by considering the properties of the coarse and fine aggregates. The aggregate was ordered in a large quantity with the factor of safety applied, and the amount calculated from the mix design was well below this limit. The reason for ordering a larger quantity of material initially was that we wished to purchase the material from the same batch so that the properties were consistent. A trial mix was prepared for the slump test (ASTM C 143). The target slump was 75–100 mm, but the trial mix yielded a slump greater than 150 mm. Concrete was produced by combining CBA fume, WGS, cement, water and fine and coarse aggregates along superplasticizer. Cement was tested using 43-grade ordinary Portland cement. Table 2 shows the properties of the fine aggregates, such as the fineness modulus, bulking modulus and specific gravity values.

### 2.2. Mixing and Casting Procedure

With the use of the design of experiments (DOE) methodology and the Design Expert software, the percentage of WGS in the modified high-volume CBA concrete was determined. According to the RSM study, the percentage of CBA varied from 0% to 20%, while the percentage of WGS varied from 0% to 20%. Table 3 shows the total combination with various amounts of CBA and WGS. Three phases of mixing were completed: three minutes of dry mixing, three minutes of wet mixing, three minutes of SP addition and at least four minutes of final mixing. Fresh concrete’s characteristics were then analyzed. Following this, the freshly mixed concrete was poured into standard molds, one measuring 100 millimeters in diameter by 200 millimeters tall by 500 millimeters wide. A hardened prism was used for the compressive strength test. For each combination, the loading rate was set at 3.0 KN/s, and the hardened prisms were verified after 28 and 90 days on average, with three samples analyzed in each case. After curing for 28 and 90 days, the splitting tensile strength of specimens was tested, as required by ASTM C293M-10.

### 2.3. Testing Program

The workability of concrete was assessed using the slump test, while the mechanical properties (compressive strength, split tensile strength, ultrasonic pulse velocity and rebound number) were examined using only one test on fresh concrete specimens. These tests were carried out in accordance with ASTM C192/C 192 M-06. The size of the prisms used was 150 mm × 150 mm × 150 mm; in addition to ordinary concrete and CBA and WGS concrete, the samples were cast. Table 4 shows the tests for measuring standards of the materials.

### 2.4. Response Surface Method

The response surface method (RSM) is a quantifiable means of creating mathematical models that show one or more responses within a series of input variables [34]. The RSM provides a polynomial relationship between the response and the input elements, calculating the impact and relevance of each. This demonstration can be used to predict and optimize the response of a mixed design. The collection of experimental data is the first step in developing a statistical model, followed by selecting an appropriate model to match the data. The demonstration at this point focuses on whether the appraisal is adequate. Design Expert v11 is a measurable computer software program that includes test plans, numerical equations, factual investigation and response optimization. The analysis of variance (ANOVA) tool is used to design the interaction between input components and their impact on the response [35]. The compressive strengths (y1) of the specimens, obtained utilizing the slump test, are the responses investigated in this study (y4). WGS/B (x1) and CBA/B (x2), which refer to the proportion of fly ash and silica fume, respectively, are the factors that govern these reactions. They individually account for the entirety of the cement’s vent gas substance.

## 3. Results and Discussion

Using both destructive and non-destructive methods, laboratory experiments were conducted on the strength and modulus of elasticity indexes of concrete. NDT and the standard compressive and modulus of elasticity tests on the same concrete specimens must be shown to be related. In the previous section, a detailed explanation of the concrete mix was presented.

### 3.1. Compressive Strength

Concrete containing varied percentages of coal bottom ash and waste glass sludge at different ages is depicted in Figure 1. Initially, control specimens were cast and the compressive strength of concrete was tested at 28 days and 90 days. The compressive strength was 39.2 and 39.4 MPa at 28 days and 90 days, respectively. The compressive strength of concrete specimens with 5% to 10% coal bottom ash and waste glass sludge was found to be higher than that of the control specimen. Concrete having 10% CBA had the highest strength of any other type of concrete. Due to the reduced agglomeration of the solid particles, the compressive strength of the material was increased. One of the possible reasons for the improved compressive results is the filling ability and the binder effect of CBA and WGS. The micro- and nano-sized particles of WGS and CBA helped to improve the porous structure of concrete, which is a heterogonous material. This resulted in an increase in the flow ability of fresh concrete and a decrease in the porosity of hardened concrete, both of which had a significant impact on the strength and durability of concrete. The findings of this investigation are congruent with the findings of previous studies [35,36,37], which found that the addition of coal bottom ash at up to 10% in concrete resulted in positive compressive strength. In the plots, the red zone signifies a higher-strength region, whereas the green region symbolizes a lower-strength region of concrete, as shown in the 2D contour plot in Figure 2b. The red zone has the maximum compressive strength value, followed by the yellow region, and the green region has the minimum compressive strength value. For concrete combined with WGS, the ideal compressive strength was 38 MPa at 10% WGS, respectively. The contours’ skewed appearance shows that there is a weak interface between the factors (percentage of WGS). According to the 3D response surface plot in Figure 2a, the compressive strength decreases dramatically as the WGS concentration increases.

Figure 3 depicts the variation in the compressive strength of all concrete mixtures when combined in various ways. The red zone signifies a higher-strength region, whereas the green region symbolizes a lower-strength region of concrete, as shown in the 2D contour plot in Figure 3a,b. The red zone has the maximum compressive strength value, followed by the yellow region, and the green region has the minimum compressive strength value. For concrete combined with CBA, the ideal compressive strength was 42.0 MPa at 10% CBA, respectively. The contours’ skewed appearance shows that there is a weak interface between the factors (percentage of CBA). According to the 3D response surface plot in Figure 3b, the compressive strength decreases dramatically as the WGS concentration increases.

### 3.2. Ultrasonic Pulse Velocity

Using ultrasonic pulse velocity equipment, we also measured the ultrasonic pulse velocity of concrete with different dosages of CBA and WGS. A minimal influence of CBA and WGS may be seen in the UPV results at 28 and 90 days after the casting of all types of concrete. At 90 days, there was a minor increase in UPV due to the addition of coal bottom ash, but at all ages, there was a decrease due to the addition of 15 percent CBA and WGS. The porosity of concrete is a major factor in the development of UPV. UPV increases as the concrete hardens due to the decrease in porosity caused by an increase in density [10,38].

When CBA and WGS are incorporated into concrete, their smaller particle sizes reduce the porosity, raising the UPV of the final product. Cement content decreases with the addition of CBA and WGS, resulting in less CSH gel and less dense concrete, which is why the UPV decreases with 15% substitution of these components, as shown in Figure 4. There are correlations between the compressive strength and the proportion of industrial waste in the concrete, as indicated in the graphs. The strength gains follow a similar pattern, according to the data. The linear trends in ultrasonic pulse velocity illustrate that the compressive strength of concrete increases with time.

### 3.3. Compressive Strength and Rebound Hammer

The results of the rebound number are represented in Figure 5. Concrete including 5–10% CBA and WGS showed higher rebound numbers than control concrete at 28 and 90 days. Despite this, the rebound number was found to be somewhat lower than that of the reference concrete when the CBA and WGS content was increased over 10%. Because the surface hardness of concrete improves as it ages and the percentage of industrial waste in concrete increases, this phenomenon can account for the cementitious effect of supplementary fillers. The amount of cement in this form of concrete is small, and as a result, the surface hardness is decreased by 15–20%, according to the CBA and WGS percentages. The graphs below show the relationships between the rebound number and the compressive strength of concrete that contains various amounts of industrial waste. The strength gains follow a similar pattern, according to the data. As the compressive strength of concrete increases with age, the rebound number also increases, according to the graph [24,39].

The rebound number results are depicted in Figure 5 at 28 and 90 days, concrete containing 5–10 % CBA and WGS rebounded more strongly than control concrete. When the CBA and WGS content increased above 10%, it resulted in a rebound number that was lower than that of the reference concrete. This phenomenon can be explained by the fact that concrete’s surface hardness increases with age and as the percentage of industrial waste in the concrete rises [12,13,14]. Based on data from CBA and WGS, the surface hardness was lowered by 15–20 percent in this type of concrete. The rebound and compressive strength of concrete containing varying amounts of industrial waste are depicted in the graphs below. According to the research, the strength gains follow a predictable pattern. According to the graph, as concrete ages, its rebound number grows in tandem with its compressive strength. For concrete strengths ranging from 20 MPa to 46 MPa, the multiple regression analysis also suggested the following relationships.

### 3.4. Anova Analysis

The effect of three independent variables, a combination of 10% CR (B), 15% SF and 10% FA, on concrete strength was investigated (A). Analysis of variance was used to establish the model’s significance (ANOVA). Table 4 illustrates the outcomes. The compressive strength for CBA and WGS at 42.1 MPa and 38.0 MPa, respectively, of the model had an F value of 13.05, as indicated in Table 5 and Table 6 A high F value denotes the model’s relevance and appropriateness. The figures show that all models are significant. The model’s F value has a 0.01 percent risk of being affected by noise. Models with *p* values less than 0.05 are considered essential. 

The coefficient of determination, often known as R-squared (R2), of the model should be near 0.88 to demonstrate its reliability. The adjusted R2 and projected R2 have a variance of less than 0.2. This suggests that the modified R2 and the projected R2 are reasonably consistent. The signal-to-noise ratio measurement is also a minimum requirement for optimal (Adeq) accuracy. All models can navigate the design space, as shown in Table 6. One technique to validate the model is to plot the residual normal plot, as a straight line. As a result, if all points are almost parallel to the normal line, the predicted value will produce more accurate results than expected [34,35]. The validity of the model and the capacity to select the appropriate extraction parameters based on the response outcomes are determined using normal residual plots.

### 3.5. Optimization and Experimental Validation

The impact of the interaction of two factors, silica fume and rubber particles, on the mechanical strength was modeled using the response surface methodology (RSM). When all of the data from the first mixture were effectively acquired, 13 different mixture designs were developed based on the first RSM mixture design and re-entered into the RSM to verify their efficacy, which offered the final model, shown in Table 7. The comparison results demonstrate that the experiment and RSM findings differ slightly, but this is still acceptable. The desirable value is 0.592, indicating that with 10% CBA and 10 WGS, 42.1 MPa and 38.0 MPa compressive strength can be achieved, respectively. 

## 4. Microstructure Analysis by SEM

### 4.1. Waste Glass Sludge (WGS)

By examining the WGS SEM images in Figure 6 crushed and angular particles can be seen. More gel-type C-S-H can be seen in WGS samples. The calcium silicate hydrate (C-S-H) is mostly in an amorphous phase or poorly crystalline, while less pores and voids can be found in WGS 10%, which also indicates its greater compressive strength due to its good pozzolanic characteristics compared to other samples, as verified above [40]. The fine glass particles can be clearly seen with the C-S-H phase. The combined phase results in a honeycomb structure. The pozzolanic activity of the WGS is mainly considered to be due to the amorphous fine particles, which react with the portlandite to expedite the C-S-H formation. Moreover, the CBA and WGS particles help in the densification and act as a filler agent [16,41].

### 4.2. Coal Bottom Ash (CBA)

Traces of needle-like ettringite content can be clearly observed in the SEM images in Figure 5. The C-S-H can also be seen, but it is not so extensive as WGS [42]. Less pores and voids can be found in CBA 10%, which may result from the good pozzolanic quality and high compressive strength. Calcium alumino-silicate hydrate (C-A-S-H) can also be found in CBA, as verified previously [42]. The portlandite phase can be seen in C-S-H, which is also verified by previous research work [43]. The available CH helps in producing more C-S-H in the concrete, which serves to improve the strength and density of the concrete [42,44].

### 4.3. Calculation of Porosity by Analyzing Particles Using Image J Software

Porosity analysis can be performed using the Image J software [16,40]. In this analysis, the SEM image is refined in the Image J software and then the image channels are divided for better visibility, and an appropriate threshold level is applied to find the particles to be analyzed. Then, we analyzed particles’ summary will indicate the required area of the pores. Figure 7 shows the threshold level applied to visualize the particles for analysis. The details for the particles are shown in Table 8. The minimum porosity was shown by CBA and WGS at 10% partial cement replacement. Thus, these two samples also showed the highest compressive strength. This might have been due to their pozzolanic behavior, which ultimately fills the voids and cracks [23]. Thus, CBA and WGS are both good replacements for cement binders.

## 5. Conclusions and Recommendations

### 5.1. Conclusions

This investigation included two phases, the first of which involved the use of destructive and non-destructive mechanisms to assess the concrete strength of prisms. Hammer and ultrasonic pulse velocity testing showed validity of the correlations.

The following conclusions were made after the experimental and analytical discussions.

The maximum increase in compressive strength was observed with 10% CBA and WGS addition after 28 days, and this can be considered the optimum dosage of coal bottom ash for normal-strength concrete.

Design Expert software was also used to assess the samples at all ages, and a similar trend was observed as that of compressive strength. RSM was also validated for the prediction of compressive strength at different ages; however, approximately 15% higher experimental values were observed at the age of 90 days.

Relationships such as ultrasonic pulse velocity–compressive strength, rebound number–compressive strength and dynamic modulus of elasticity–static modulus of elasticity were proposed in this work for all percentages (5%, 10%, 15% and 20%) of CBA and WGS.

The combined use of pundit and rebound hammer as non-destructive test methods for the assessment of both early-age and aging concrete gave more desirable results than each method used in isolation.

General relationships were also proposed for CBA- and WGS-incorporated concrete for strengths ranging from 20 MPa to 46 MPa. The coefficients of determination for these relationships were found to be less than 90% (0.85, 0.82), which was due to the larger number of variables used to obtain these relationships.

To determine whether the predictive models were statistically significant, we utilized a two-way ANOVA with a significance criterion of a *p* value less than 0.05. An equation was proposed to relate the compressive strength of CBA and WGS concrete. The RSM model and the equations utilized to calculate CBA and WGS concrete’s compressive strength were accurate. The prediction model results validated the experimental data, revealing a slight variance.

Based on the results of this research and previous research, the utilization of coal bottom ash and waste glass sludge as partial substitutions for cement in concrete is highly recommended as they do not only provide improved mechanical properties but also offer suitable solutions to achieve sustainable and environmentally friendly processes.

### 5.2. Recommendations

The research findings in this work demonstrate that ultrasonic pulse velocity appears more reliable in predicting the possible compressive strength of concrete, especially for aging concrete, which is the major focus of structural health monitoring in concrete structures. However, the results should be taken with caution and confirmed with proven reliable calibration(s). The results show that pundit values offer higher statistical confidence than the rebound number. Based on the results, one may recommend that coal bottom ash and waste glass sludge can effectively be utilized as partial substitution for cement.

## Figures and Tables

**Figure 1 materials-15-07346-f001:**
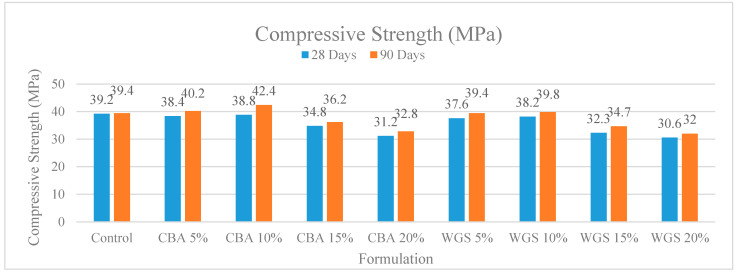
Comparison chart of compressive strength of concrete at 28 and 90 days.

**Figure 2 materials-15-07346-f002:**
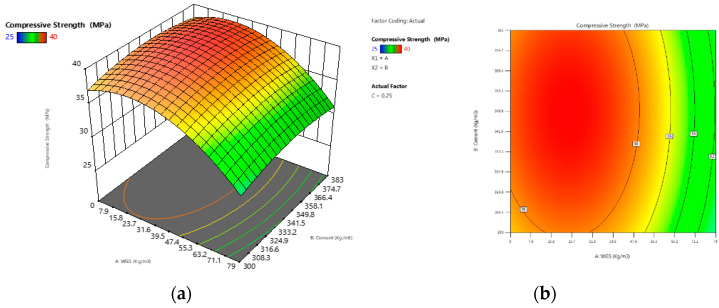
(**a**) The 3D surface diagram for compressive strength (WGS); (**b**) the 2D contour for compressive strength (WGS).

**Figure 3 materials-15-07346-f003:**
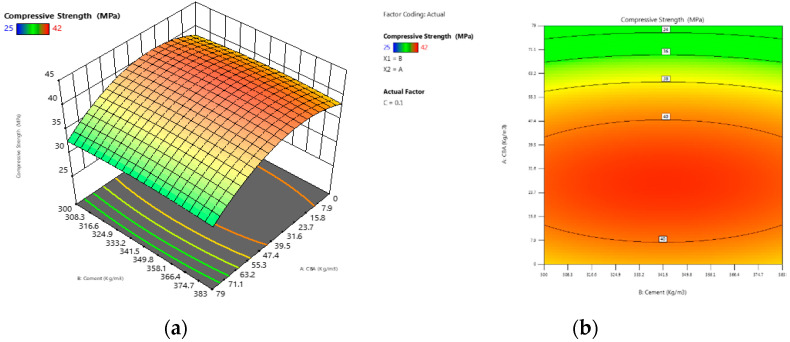
(**a**) The 3D surface diagram for compressive strength (CBA); (**b**) the 2D contour for compressive strength (CBA).

**Figure 4 materials-15-07346-f004:**
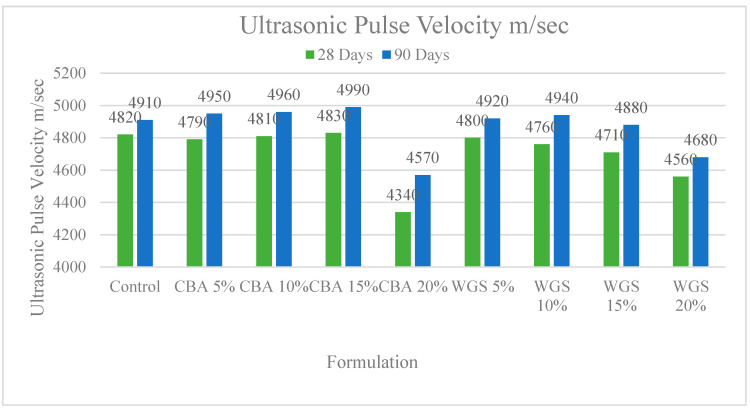
Comparison chart of ultrasonic pulse velocity of concrete at 28 and 90 days.

**Figure 5 materials-15-07346-f005:**
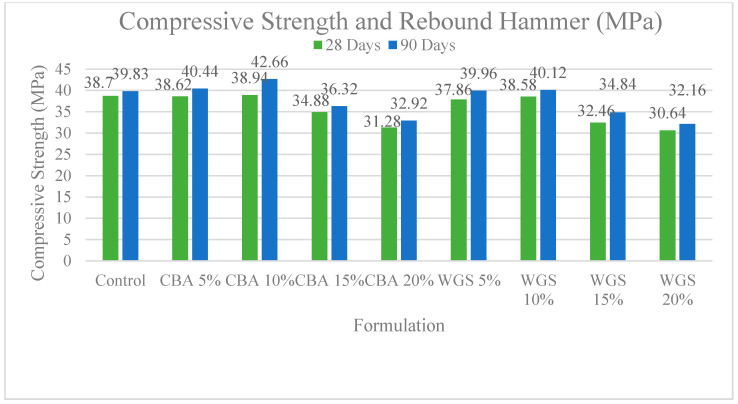
Comparison chart of rebound number of concrete at 28 and 90 days.

**Figure 6 materials-15-07346-f006:**
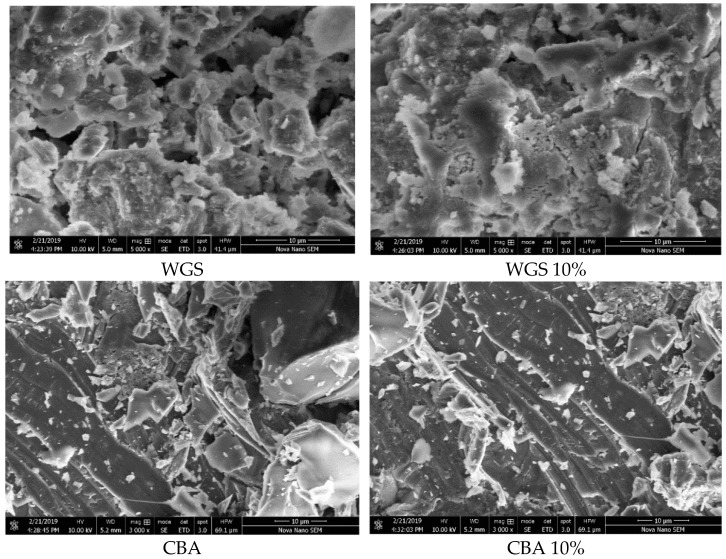
SEM analysis of WGS & CBA.

**Figure 7 materials-15-07346-f007:**
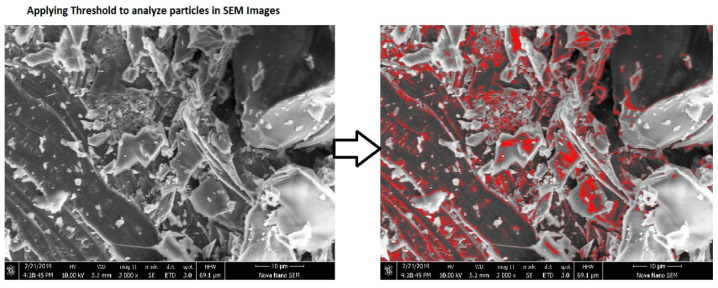
Threshold level applied to visualize the particles.

**Table 1 materials-15-07346-t001:** Chemical properties of cement, CBA and WGS.

Oxide (%)	OPC Type 1	CBA	WGS
CaO	65	5.25	9.94
SiO_2_	17.15	75	66.6
Na_2_O	1.86	0.32	7.56
MgO	1.74	1.4	2.92
Al_2_O_3_	5.6	25.2	10.1
K_2_O	1.19	2.52	0.219
Fe_2_O_3_	3.21	9.05	0.226
SO_3_	2.66	1.6	0.264
TiO_2_	0.32	-	
P. Size, μm	±20	13	

**Table 2 materials-15-07346-t002:** Properties of aggregates.

Property	Fine Aggregate	Course Aggregate	Range	Standard
Aggregate Particle Size	Less than 4.75 mm	20 mm nominal size	4.75 mm to 20 mm	ASTM: C192/C 192M-06
Specific Gravity	2.65	2.72	2.5 mm to 3.0 mm	AASHTO T 85-88
Fineness Modulus	2.7	4.3	2.5 to 5.0	AASHTO T 27

**Table 3 materials-15-07346-t003:** Defining cases for each mix design sample.

Nomenclature	Cement kg/m^3^	Fine Aggregate kg/m^3^	Coarse Aggregate kg/m^3^	Water kg/m^3^	WGS kg/m^3^	CBA kg/m^3^	Superplasticizer kg/m^3^
C1	383	467	1034	160	0	0	4.5
C1-WGS 5	363.9	467	1034	160	19.1	0	4.5
C1- WGS 10	344.8	467	1034	160	38.3	0	4.5
C1- WGS 15	325.7	467	1034	160	57.3	0	4.5
C1- WGS 20	306.6	467	1034	160	78.6	0	4.5
C1-CBA 5	363.9	467	1034	160	0	19.1	4.5
C1-CBA 10	344.8	467	1034	160	0	38.3	4.5
C1-CBA 10	325.7	467	1034	160	0	57.3	4.5
C1-CBA 10	306.6	467	1034	160	0	78.6	4.5

**Table 4 materials-15-07346-t004:** Tests for measuring standards of the materials.

Sr.	Test	Code
1	Soundness Test of Cement	(ASTM C187-16)
2	Setting Time of Cement	(ASTM C191-13)
3	Fineness Modulus of Sand	(ASTM C136)
4	Abrasion Value of Coarse Aggregates	(ASTM C 535)
5	Crushing Strength & Impact Value of Coarse Aggregates	ASTM D7137/D7137M-17
6	Flakiness, Elongation & Angularity No. of Course Aggregates	(ASTM D 4791)
7	Specific Gravity & Water Absorption of Coarse Aggregates	(AASHTO T 85-88)
8	Slump Test	(ASTM C143)
9	Compression Test	(ATSM C39)
11	Scanning Electron Microscopy (SEM)	ASTM C1723-16.
12	Response Surface Methodology (RSM)	Design Expert

**Table 5 materials-15-07346-t005:** ANOVA analysis of the response models.

Source	Sum of Squares	df	Mean Square	F Value	*p* Value	
Model	326.13	5	65.23	13.5	0.0004	significant
A-WGS	158.33	1	158.33	32.77	0.0002	
B-Cement	3.31	1	3.31	0.686	0.4269	
AB	0	1	0	0	1	
A^2^	164.47	1	164.47	34.05	0.0002	
B^2^	15.42	1	15.42	3.19	0.1043	
Residual	48.31	10	4.83			
Lack of Fit	48.31	3	16.1			
Pure Error	0	7	0			
Cor Total	374.44	15				
Model	361.04	5	72.21	9.71	0.0014	significant
A-CBA	108.53	1	108.53	14.59	0.0034	
B-Cement	5.68 × 10^−14^	1	5.68 × 10^−14^	7.64 × 10^−15^	1	
AB	0	1	0	0	1	
A^2^	244.92	1	244.92	32.92	0.0002	
B^2^	4.03	1	4.03	0.5413	0.4788	
Residual	74.4	10	7.44			
Lack of Fit	74.4	3	24.8			
Pure Error	0	7	0			
Cor Total	435.44	15				

**Table 6 materials-15-07346-t006:** Models’ validation.

Source	Sequential *p* Value	Model Terms	Compressive Strength WGS	Compressive Strength CBA
Quadratic model	<0.0001	Std. Dev	2.40	2.83
Quadratic model	<0.0001	Mean	37.0	39.6
Quadratic model	<0.0001	C.V %	7.12	7.82
Quadratic model	<0.0001	R2	0.8710	0.8491
Quadratic model	<0.0001	Adj. R2	0.8265	0.8137
Quadratic model	<0.0001	Pred. R2	0.8046	0.7923

**Table 7 materials-15-07346-t007:** Summary of optimization of mix.

No. of Runs	WGS	CBA	Compressive Strength (MPa)	Compressive Strength (MPa)
1	10	0	38	
2	10	0	38.1	
3	10	0	38.08	
4	0	10		42.1
5	0	10		42.14
6	0	10		42.0
7	0	0	34	34
Mean			37.04	40.06

**Table 8 materials-15-07346-t008:** Analysis of pores by Image J.

Sample Slice	Count	Total Area	Average Size	% Area
CBA	9036	30.957	0.003	7.748
CBA 10%	5435	12.131	0.002	3.228
WGS	554	23.097	0.042	5.650
WGS 10%	844	11.726	0.014	3.523

## Data Availability

The data are available on request to the corresponding author.

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
