# Peer review of "An Experimental Study on Non-Destructive Evaluation of the Mechanical Characteristics of a Sustainable Concrete Incorporating Industrial Waste"

_materials, 2022, doi:10.3390/ma15207346_

Round 1
Reviewer 1 Report
This article investigates the mechanical characteristics of high-strength concrete comprising two different industrial wastes (coal bottom ash and waste glass sludge). The percentage of coal bottom ash varied from 0% to 20%, while the percentage of waste glass sludge varied from 0% to 20%. The non-destructive test methods were used to evaluate the compressive strength and quality of concrete. This article needs some improvement before it can be accepted for publication.
1. Line 22, define the acronym “NDT” when used the first time.
2. I suggest modifying the sentences “There were 162 prisms, each 150x150x150 mm in size, composed of normal 26 strength concrete, with varying percentages of industrial waste being used in their production”.
3. Abstract: The description of the work investigated is unclear and it should be improved by adding more details.
4. The citation must be presented when using a statistic “accounting for 76% of the total-line46”.
5. The acronyms have been defined in multiple locations “for example, line, 101, 104, 110 etc. Please avoid repetitions.
6. The literature section is lacking in discussing the mechanical properties of concrete comprising industrial waste. Please add more literature relevant to this.
7. Highlight the novelty of the research at the end of the introduction section.
8. Materials and methods: it is recommended to describe the materials used one by one. What are the properties of cement (setting time, specific gravity, consistency etc)?. Properties of fine and coarse aggregate?, Granulometric curves for the fine and coarse aggregate?. Superplasticizer used, if any?. This section is shallow with fewer details and it should be improved. Please introduce a new section to the materials used and discuss everything under this section.
9. Why did the CBA percentage vary from 0% to 20%, while the percentage of WGS varied from 0% to 20%?. Please highlight the reason for this selection.
10. More details about the WGA could be included in the manuscript. Is the treated or raw WGA used in the study?
11. Why was superplasticizer constant for all the mixtures while increasing the content of WGA and CBA?. These materials do not influence workability?.
12. Table 4, no discussions and results are presented based. What is the rationale for Table 4?
13. Please highlight the limitations of test data used for the Response Surface Method.
14. The compressive strength increment for each mixture should be compared with the control mixture. The discussion is lacking in this aspect.
15. How to ensure the results' accuracy in Table 7?
16. It is recommended to add a few quantifying results for compressive strength.
Author Response
Detailed Response to Editors and Reviewers
Manuscript submitted to Materials: materials-1979894, entitled "An experimental study on non-destructive evaluation of the mechanical characteristics of high strength concrete incorporating industrial waste.
Dear Editor Abby Wang,
We appreciate the valuable time that you and all the reviewers have taken to review this paper. We have accommodated these review comments as follow:
1. Responses to the reviewer # 1’s comments
1.1 General Comments:
This article investigates the mechanical characteristics of high-strength concrete comprising two different industrial wastes (coal bottom ash and waste glass sludge). The percentage of coal bottom ash varied from 0% to 20%, while the percentage of waste glass sludge varied from 0% to 20%. The non-destructive test methods were used to evaluate the compressive strength and quality of concrete. This article needs some improvement before it can be accepted for publication.
Response
The authors would like to thank whole-heartedly Reviewer # 1 for the detailed review of the paper, critical comments and valuable suggestions for improving this work. The authors have carefully addressed the reviewer’s specific points. I hope that our revision will make the manuscript meet the publication requirements.
- Specific Comments
[1] Line 22, define the acronym “NDT” when used the first time.
Response
It is a right observation made by a respected reviewer. The acronym for NDT is added in the revised version.
[2] I suggest modifying the sentences “There were 162 prisms, each 150x150x150 mm in size, composed of normal 26 strength concrete, with varying percentages of industrial waste being used in their production”.
Varying quantity of Industrial waste was incorporated to cast concrete prisms (150mm x 150mm x 150mm)
Response
Authors would like to thank the reviewer for this valuable comment. The lines have been rephrased in the revised manuscript file.
[3] Abstract: The description of the work investigated is unclear and it should be improved by adding more details.
Response
The reviewer has raised a very important point here.
[4] The citation must be presented when using a statistic “accounting for 76% of the total-line46”.
Response
The authors would like to thank Reviewer #1 for the important comment. The authors have added the relevant citation in the revised version.
[5] The acronyms have been defined in multiple locations “for example, line, 101, 104, 110 etc. Please avoid repetitions.
Response
The authors would like to thank Reviewer #1 for pointing out this mistake. The authors have deleted the repeated words in the new version of the manuscript.
[6] The literature section is lacking in discussing the mechanical properties of concrete comprising industrial waste. Please add more literature relevant to this.
Response
The authors are grateful to the reviewer for the critical comment. New literature has been added in the revised manuscript.
[7] Highlight the novelty of the research at the end of the introduction section.
Response
The authors realize that it is an important comment from the reviewer. The novelty part is added at the part of the introduction of new version.
[8] Materials and methods: it is recommended to describe the materials used one by one. What are the properties of cement (setting time, specific gravity, consistency etc.)? Properties of fine and coarse aggregate? Granulometric curves for the fine and coarse aggregate? Superplasticizer used, if any? This section is shallow with fewer details and it should be improved. Please introduce a new section to the materials used and discuss everything under this section.
Response
The authors are grateful to the reviewer for pointing out this important missing point. The authors have added the following part in the Materials and Mixing part (2.1)
Ordinary Portland cement was used in this experimental campaign. Chemical composition of cement is shown in the table 1. CBA and WGS were also incorporated as a binder material. Detailed composition is shown in the table 1. Fine aggregates having a fineness modulus of 2.7 was used while a coarse aggregate with a fineness modulus of 4.3 was incorporated. Properties of used aggregates is shown in the table 2. Polycarboxylate based superplasticizer was used as a concrete admixture to maintain the workability.
[9] Why did the CBA percentage vary from 0% to 20%, while the percentage of WGS varied from 0% to 20%. Please highlight the reason for this selection. Kindly add a table that describes the main physical and chemical properties of the raw materials used in this study.
Response
The authors want to thank the reviewer for this technical question. The main reason to change the percentage from 0% to 20% was to have an understanding of the waste material’s influence on binding effect of concrete. In this regard the percentage was raised from 0% to 20% to get an ideal percentage of cement replacement. Besides, a table is added (Table#1) to present the chemical composition of CBA & WGS.
[10] More details about the WGA could be included in the manuscript. Is the treated or raw WGA used in the study?
Response
The details on chemical composition is added in the manuscript. The authors have added the information on WGS mentioning that WGS used was in fine powder form which is not been to any kind of pretreatment.
[11] Why was superplasticizer constant for all the mixtures while increasing the content of WGA and CBA? These materials do not influence workability? Were the preparation methods for mixing and casting that were described by the authors come in accordance with a certain standard or do they follow previous procedures.
Response
The reviewer has raised a very important and convincing point here. The workability of the industrial waste always remains a concern. However, in our scenario the replacement was partial. For test cases we checked the workability of concrete specimens at higher dosage of CBA and WGS. We noticed a big change in workability and it also affected the strength of concrete as well. Therefore, we chose a concrete with minimum adverse effects on workability and strength.
The method used for mixing and casting used in this is based on the experimental is based on the previous research and experimental work experience of the authors.
Table 4, no discussions and results are presented based. What is the rationale for Table 4? For the compressive tests, what was the reason for the specified test conditions in this research? Do the speed of test value relate to a specific application?
[12] Table 4, no discussions and results are presented based. What is the rationale for Table 4? For the compressive tests, what was the reason for the specified test conditions in this research? Do the speed of test value relate to a specific application?
Response
Some of the results and discussion mentioned in table 4 are not discussed here. For instance, Crushing Strength & impact value of coarse aggregates, flakiness index etc. We performed these tests for our own understandings and to know if the gradation is correct. However, if the reviewer suggests to remove the table we can delete the table in the final version. The speed of test value is based on the normal recommendation and our previous research practices.
[13] Please highlight the limitations of test data used for the Response Surface Method.
Response
Authors would like to thank the reviewer for this observation. The proposed data provided by this software is quite similar to the test results. However, as a matter of fact It is recommended to counter verify the proposed results provided by RSM through experimental validation. One of the limitation of this software that we realize is the need of large data. This data is difficulty to validate through experimental results. Therefore, the authors chose the data nearest to the nominal design strength values.
[14] The compressive strength increment for each mixture should be compared with the control mixture. The discussion is lacking in this aspect. Regarding to the compression strength at 28 days and 90 days, why do some samples don't show any significant change in the strength while other samples showed some variation?
Response
Authors realized that the point and observation raised by the reviewer is very valid and important. In order to link the test results of the experimental program the authors have added few lines on the control specimens. Besides, the authors would like to elaborate the reason of improving the strength in certain cases while the strength remains almost unchanged in certain cases. As we had used different percentages of industrial waste and the main reason of this study was to find a suitable dosage that would help to achieve maximum outcomes and it would also be helpful in minimizing the cement consumption which is a great cause to the global emissions.
- How to ensure the results' accuracy in Table 7?
Response
The results presented in table 7 are quite similar to the experimental results which shows its validation. Besides, the 3D graphs presented show the correlation with the experimental results and there’s a little difference.
- It is recommended to add a few quantifying results for compressive strength.
Response
The authors would like to thank for this valuable comment. Some new commentary on the reasons of increment in compressive strength is added in the revised manuscript in line 218 for the kind consideration.
Reviewer 2 Report
The paper presents an interesting approach based on the experimental study on non-destructive evaluation of the mechanical characteristics of high strength concrete incorporating industrial waste. However, the innovation of the current research work should be further highlighted and emphasized. At the same time, the authors should consider the following comments to greatly improve the quality of the paper.
1. In the abstract, add a final statement that highlights the importance of this research and its possible potentials.
2. The introduction needs to be improved by relating to the mechanics of the studied materials and their mechanical characteristics. The references to be included are: 10.1177/0021998318790093, 10.1016/j.polymertesting.2017.09.009, 10.1016/j.compstruct.2021.114698, 10.1177/0731684417727143, 10.1002/app.46770, 10.1016/j.porgcoat.2022.107015.
3. Kindly add a table that describes the main physical and chemical properties of the raw materials used in this study.
4. Were the preparation methods for mixing and casting that were described by the authors come in accordance with a certain standard or do they follow previous procedures?
5. For the compressive tests, what was the reason for the specified test conditions in this research? Do the speed of test value relate to a specific application?
6. How many samples were used per configuration for the compression test?
7. Regarding to the compression strength at 28 days and 90 days, why do some samples don't show any significant change in the strength while other samples showed some variation?
8. The conclusion is too long. It needs to be modified to summarize the research outcomes in short statements with clear observations.
Author Response
Detailed Response to Editors and Reviewers
Manuscript submitted to Materials: materials-1979894, entitled "An experimental study on non-destructive evaluation of the mechanical characteristics of high strength concrete incorporating industrial waste.
Dear Editor Abby Wang,
We appreciate the valuable time that you and all the reviewers have taken to review this paper. We have accommodated these review comments as follow:
- Responses to the reviewer # 2’s comments
1.1 General Comments:
The paper presents an interesting approach based on the experimental study on non-destructive evaluation of the mechanical characteristics of high strength concrete incorporating industrial waste. However, the innovation of the current research work should be further highlighted and emphasized. At the same time, the authors should consider the following comments to greatly improve the quality of the paper.
Response
The authors would like to thank whole-heartedly Reviewer # 2 for the detailed review of the paper, critical comments and valuable suggestions for improving this work. The authors have written a specific paragraph on the novelty of this research work at the end of introduction section. The authors have carefully addressed the reviewer’s specific points. I hope that our revision will make the manuscript meet the publication requirements.
[1]. In the abstract, add a final statement that highlights the importance of this research and its possible potentials.
Response
This is a right and important observation raised by the respected reviewer. The abstract is substantially improved in the revised version of the manuscript for the kind competent consideration of the reviewer.
[2]. The introduction needs to be improved by relating to the mechanics of the studied materials and their mechanical characteristics. The references to be included are: 10.1177/0021998318790093,
10.1016/j.polymertesting.2017.09.009,
10.1016/j.compstruct.2021.114698,
10.1177/0731684417727143,
10.1002/app.46770, 10.1016/j.porgcoat.2022.107015.
The authors want to thank the competent reviewer for pointing out the lacking in this important area of paper. In this regard the authors have added the relevant studies mentioned by the reviewer. The authors once again thank the reviewer to bring our attention towards this important point.
[3]. Kindly add a table that describes the main physical and chemical properties of the raw materials used in this study.
Response
The authors have added the chemical composition of cement, CBA, and WGS in table 1 for the consideration of reviewer.
[4]. How many samples were used per configuration for the compression test?
Response
It is worth mentioning that at least 4 specimens were tested for each case and an average was taken as compressive strength.
[5]. Regarding to the compression strength at 28 days and 90 days, why do some samples don't show any significant change in the strength while other samples showed some variation?
Response
The authors would like to thank for a very technical comment on this important point. The authors had the same observations regarding the little improvement on compressive strength of concrete. However, the authors concluded that one of the possible reasons could be the influence of the addition of the industrials wastes and the partial substitution of cement. The authors intended to carry a study in future the long term mechanical and durability properties of the concrete with similar composition.
[6]. Were the preparation methods for mixing and casting that were described by the authors come in accordance with a certain standard or do they follow previous procedures?
Response
The method used for mixing and casting used in this is based on the experimental is based on the previous research and experimental work experience of the authors. It is worth mentioning that it doesn’t affect the mechanical properties of concrete.
[7]. For the compressive tests, what was the reason for the specified test conditions in this research? Do the speed of test value relate to a specific application?
Response
The specified tests conditions were used in our previous research works. Besides, the speed of test value is based on the normal recommendation and our previous practices.
[8]. The conclusion is too long. It needs to be modified to summarize the research outcomes in short statements with clear observations.
Response
The authors would like to thank the reviewer for this valuable observation. The authors have removed the extra information provided in the revised version.
Reviewer 3 Report
The authors conducted destructive and non-destructive tests on the concrete with industrial wastes.
There are major issues before publication:
Can a concrete less than 40 MPa called as high strength?
The importance of using recycled materials and wastes on sustainable concrete should more emphasized in the introduction. Following studies should be used for this purpose: Improvement in Bending Performance of Reinforced Concrete Beams Produced with Waste Lathe Scraps; Concrete Containing Waste Glass as An Environmentally Friendly Aggregate: A Review on Fresh and Mechanical Characteristics; Performance assessment of fiber-reinforced concrete produced with waste lathe fibers
More information should be provided on the tests
How can CBA increase capacity?
How many tests were conducted for each mix?
What is the standard deviaiton?
More detail should be provided for SEM
Author Response
Responses to the reviewer # 3’s comments
The authors would like to thank Reviewer #3 for the critical comments and the valuable time to review the manuscript during his/her busy schedule. The authors have carefully addressed the reviewer’s general and specific comments.
General Comments
[1] The authors conducted destructive and non-destructive tests on the concrete with industrial wastes.
There are major issues before publication:
Can a concrete less than 40 MPa called as high strength?
Response
Accepted, the authors highly admire the highly valuable comments of the reviewer. The authors second the observation regarding the concrete that if it is a high strength concrete. The concrete in general is called high strength concrete when the difference between the normal and tailored concrete differs a lot. We named it as high strength concrete since a sustainable and environment concrete was designed using industrial waste. The strength of this concrete is more than the normal concrete based on cement.
However, considering the valuable suggestion and observation of the reviewer the authors have changed the title of the manuscript by replacing the word high strength with Sustainable Concrete.
Specific Comments
[1] The importance of using recycled materials and wastes on sustainable concrete should more emphasized in the introduction. Following studies should be used for this purpose:
- Improvement in Bending Performance of Reinforced Concrete Beams Produced with Waste Lathe Scraps
- Concrete Containing Waste Glass as an Environmentally Friendly Aggregate: A Review on Fresh and Mechanical Characteristics
- Performance assessment of fiber-reinforced concrete produced with waste lathe fibers
Response
Accepted, the reviewer has made some important observations regarding the relevant literature especially on the concrete acquired from the industrial wastes.
The authors have added the relevant literature recommended by the eminent reviewer in the revised manuscript for the kind consideration.
[2]. More information should be provided on the tests
Response
Appreciations for this important comment. The authors have added new information on materials used to better visualize the testing of this experimental program.
[3]. How can CBA increase capacity?
Response
The authors would like to acknowledge the reviewer for this valuable comment. CBA has the cementitious properties followed by the reinforcing effects of WGS. These materials help in improving both the overall binder effect as well as the pore structure. In this regard the necessary commentary is added in the revised manuscript.
[4]. How many tests were conducted for each mix?
Response
To study each case, the authors performed 4 tests for each followed with an average of the test results which helped to improve the validation and conformity of the test results.
[5]. More detail should be provided for SEM
Response
The authors would like to thank the reviewer for this valuable and technical comment. More discussion is added in the revised manuscript for the consideration of eminent reviewer.
Round 2
Reviewer 1 Report
All comments were addressed adequately.
Reviewer 3 Report
The authors carried out the corrections.
The paper can be accepted.